# Robust high-dimensional memory-augmented neural networks

Geethan Karunaratne [1,2,3], Manuel Schmuck[1,2,3], Manuel Le Gallo [1], Giovanni Cherubini [1], Luca Benini[2], Abu Sebastian [1✉] & Abbas Rahimi [1✉]

Traditional neural networks require enormous amounts of data to build their complex mappings during a slow training procedure that hinders their abilities for relearning and adapting to new data. Memory-augmented neural networks enhance neural networks with an explicit memory to overcome these issues. Access to this explicit memory, however, occurs via soft read and write operations involving every individual memory entry, resulting in a bottleneck when implemented using the conventional von Neumann computer architecture. To overcome this bottleneck, we propose a robust architecture that employs a computational memory unit as the explicit memory performing analog in-memory computation on high-dimensional (HD) vectors, while closely matching 32-bit software-equivalent accuracy. This is achieved by a content-based attention mechanism that represents unrelated items in the computational memory with uncorrelated HD vectors, whose real-valued components can be readily approximated by binary, or bipolar components. Experimental results demonstrate the efficacy of our approach on few-shot image classification tasks on the Omniglot dataset using more than 256,000 phase-change memory devices. Our approach effectively merges the richness of deep neural network representations with HD computing that paves the way for robust vector-symbolic manipulations applicable in reasoning, fusion, and compression.

[1] IBM Research – Zurich, Rüschlikon, Switzerland. [2] Department of Information Technology and Electrical Engineering, ETH Zürich, Zürich, Switzerland. [3]These authors contributed equally: Geethan Karunaratne, Manuel Schmuck. ✉email: ase@zurich.ibm.com; abr@zurich.ibm.com

Recurrent neural networks are able to learn and perform transformations of data over extended periods of time that make them Turing-Complete[1]. However, the intrinsic memory of a recurrent neural network is stored in the vector of hidden activations and this could lead to catastrophic forgetting[2]. Moreover, the number of weights and hence the computational cost grows exponentially with memory size. To overcome this limitation, several memory-augmented neural network (MANN) architectures were proposed in recent years[3–7] that separate the information processing from memory storage.

What the MANN architectures have in common is a controller, which is a recurrent or feedforward neural network model, followed by a structured memory as an explicit memory. The controller can write to, and read from the explicit memory that is implemented as a content addressable memory (CAM), also called associative memory in many architectures[3,4,8]. Therefore, new information can be offloaded to the explicit memory, where it does not endanger the previously learned information to be overwritten subject to its memory capacity. This feature enables one-/few-shot learning, where new concepts can be rapidly assimilated from a few training examples of never-seen-before classes to be written in the explicit memory[6]. The CAM in MANN architectures is composed of a key memory (for storing and comparing learned patterns) and a value memory (for storing labels) that are jointly referred to as a key-value memory[5].

The entries in the key memory are not accessed by stating a discrete address, but by comparing a query from the controller's side with all entries. This means that access to the key memory occurs via soft read and write operations, which involve every individual memory entry instead of a single discrete entry. Between the controller and the key memory there is a content-based attention mechanism that computes a similarity score for each memory entry with respect to a given query, followed by sharpening and normalization functions. The resulting attention vector serves to read out the value memory[3]. This may lead to extremely memory intensive operations contributing to 80% of execution time[9], quickly forming a bottleneck when implemented in conventional von Neumann architectures (e.g., CPUs and GPUs), especially for tasks demanding thousands to millions of memory entries[4,10]. Moreover, complementary metal-oxide-semiconductor (CMOS) implementation of key memories is affected by leakage, area, and volatility issues, limiting their capabilities for lifelong learning[11].

One promising alternative is to realize a key memory with non-volatile memory (NVM) devices that can also serve as computational memory to efficiently execute such memory intensive operations[10,11]. Initial simulation results have suggested key memory architectures using NVM devices such as spintronic devices[10], resistive random access memory (RRAM)[12], and ferroelectric field-effect transistors (FeFETs)[13]. To map a vector component in the key memory, devices have either been simulated with high multibit precision[10], or multiple ternary CAM (TCAM) cells by using intermediate mapping functions and encoding to get a binary code[13,14]. Besides these simulation results, a recent prototype has demonstrated the use of a very small scale $2 \times 2$ TCAM array based on FeFETs[11].

However, the use of TCAM limits such architectures in many aspects. First, TCAM arrays find an exact match between the query vector and the key memory entries, or in the best case can compute the degree of match up to very few bits (i.e., limited-precision Hamming distance)[11,15], which fundamentally restricts the precision of the search. Secondly, a TCAM cannot support widely used metrics such as cosine distance. Thirdly, a TCAM is mainly used for binary classification tasks[16], because it only finds the first-nearest neighbor (i.e., the minimum Hamming distance), which degrades its performance for few-shot learning, where the similarities of a set of intra-class memory entries should be combined. Furthermore, a key challenge associated with using NVM devices and in-memory computing is the low computational precision resulting from the intrinsic randomness and device variability[17]. Hence there is need for learned representations that can be systematically transformed to robust bipolar/binary vectors at the interface of controller and key memory, for efficient inference as well as operation at scale on NVM-based hardware.

One viable option is to exploit robust binary vector representations in the key memory as used in high-dimensional (HD) computing[18], also known as vector-symbolic architectures[19]. This emerging computing paradigm postulates the generation, manipulation, and comparison of symbols represented by wide vectors that take inspiration from attributes of brain circuits including high-dimensionality and fully distributed holographic representation. When the dimensionality is in the thousands, (pseudo)randomly generated vectors are virtually orthogonal to each other with very high probability[20]. This leads to inherently robust and efficient behavior tailor-made for RRAM[21] and phase-change memory (PCM)[22] devices operating at low signal-to-noise ratio conditions. Further, the disentanglement of information encoding and memory storage is at the core of HD computing that facilitates rapid and lifelong learning[18–20]. According to this paradigm, for a given classification task, generation and manipulation of the vectors are done in an encoder designed using HD algebraic operations to correspond closely with the task of interest, whereas storage and comparison of the vectors is done with an associative memory[18]. Instead, in this work, we provide a methodology to substitute the process of designing a customized encoder with an end-to-end training of a deep neural network such that it can be coupled, as a controller, with a robust associative memory.

In the proposed algorithmic-hardware codesign approach, first, we propose a differentiable MANN architecture including a deep neural network controller that is adapted to conform with the HD computing paradigm for generating robust vectors to interface with the key memory. More specifically, a novel attention mechanism guides the powerful representation capabilities of our controller to store unrelated items in the key memory as uncorrelated HD vectors. Secondly, we propose approximations and transformations to instantiate a hardware-friendly architecture from our differentiable architecture for solving few-shot learning problems with a bipolar/binary key memory implemented as a computational memory. Finally, we verify the integrated inference functionality of the architecture through large-scale mixed hardware/software experiments, in which for the first time the largest Omniglot problem (100-way 5-shot) is established, and efficiently mapped on 256,000 PCM devices performing analog in-memory computation on 512-bit vectors.

## Results

**Proposed MANN architecture using in-memory computing**. In the MANN architectures, the key-value memory remains mostly independent of the task and input type, while the controller should be fitted to the task and especially the input type. Convolutional neural networks (CNNs) are excellent controllers for few-shot Omniglot[23] image classification task (see Methods) that has established itself as the core benchmark for the MANNs[6,11,13,14]. We have chosen a 5-layer CNN controller that provides an embedding function to map the input image to an internal feature representation (see Methods). Our central contribution is to direct the CNN controller to encode images in a way that combines the richness of deep neural network representations with robust vector-symbolic manipulations of HD

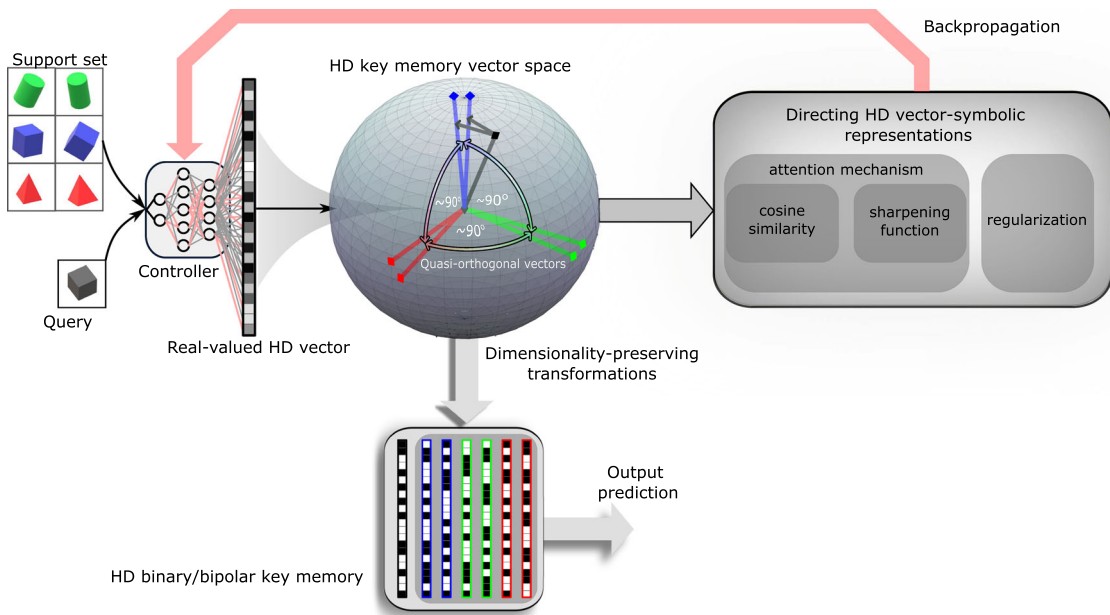

**Fig. 1 Proposed methodology to merge deep network representations with vector-symbolic representations in high-dimensional (HD) computing.** The goal is to guide a deep network controller to conform with HD computing by assigning quasi-orthogonal HD vectors to unrelated objects in the key memory. The HD vectors are then directed by adjusting the controller weights during meta-learning in such a way that the query vector gets near the set of correct class vectors, and the vectors from different classes move away from each other to produce mutually quasi-orthogonal vectors in the key memory (demonstrated in a 3D space for sake of visualization). This is achieved by using proper similarity and sharpening functions, regularizer, and expanding the vector dimensionality at the interface of controller and key memory. Then, the resulting real-valued representations can be readily transformed to dense binary/bipolar HD vectors for efficient and robust inference in a key memory using in-memory computing.

computing. Fig. 1 illustrates the steps in our methodology to achieve this goal, that is, having the CNN controller to assign quasi-orthogonal HD dense binary vectors to unrelated items in the key memory. In the first step, our methodology defines the proper choice of an attention mechanism, i.e., similarity metric and sharpening function, to enforce quasi-orthogonality (see Section "A new attention mechanism appropriate for HD geometry"). Next step is tuning the dimensionality of HD vectors between the last layer of CNN controller and the key memory. Finally, to ease inference, a set of transformation and approximation methods convert the real-valued HD vectors to the dense binary vectors (see Sections "Bipolar key memory: transforming real-valued HD vectors to bipolar" and "Binary key memory: transforming bipolar HD vectors to binary"). Or even more accurately, such binary vectors can be directly learned by applying our proposed regularization term (see Supplementary Note 1).

Our proposed MANN architecture is schematically depicted in Fig. 2. In the learning phase, our methodology trains the CNN controller to encode complex image inputs to vectors conforming with the HD computing properties. These properties encourage assigning dissimilar images to quasi-orthogonal (i.e., uncorrelated) HD vectors that can be stored, or compared with vectors already stored in an associative memory as the key-value memory with extreme robustness. Our methodology enables both controller and key-value memory to be optimized with the gradient descent methods by using differentiable similarity and sharpening functions at the interface of memory and controller. It also uses an episodic training procedure for the CNN by solving various few-shot problem sets that gradually enhance the quality of the mapping by exploiting classification errors (see the learning phase in Fig. 2). Those errors are represented as a loss, which is propagated all the way back to the controller, whose parameters are then updated to counter this loss and to reach maturity; a regularizer can be considered to closely tune the desired distribution of HD vectors. In this supervised step, the controller

is updated by learning from its own mistakes (also referred to as meta-learning). The controller finally learns to discern different image classes, mapping them far away from each other in the HD feature space.

The inference phase comprises both giving the model a few examples—that are mutually exclusive from the classes that were presented during the learning phase—to learn from, and inferring an answer with respect to those examples. During this phase, when a never-seen-before image is encountered, the controller quickly encodes and updates the key-value memory contents, that can be later retrieved for classification. This avoids relearning controller parameters through otherwise expensive and iterative training (see Supplementary Note 2). While our architecture is kept continuous to avoid violating the differentiability during the learning phase, it is simplified for the inference phase by applying transformations and approximations to derive a hardware-friendly version. These transformations directly modify the real-valued HD vectors to nearly equiprobable binary or bipolar HD vectors to be used in the key memory with memristive devices. The approximations further simplify the similarity and sharpening functions for inference (see the inference phase in Fig. 2). As a result, the similarity search is efficiently computed as the dot product by exploiting Kirchhoff's circuit laws in $\mathcal{O}(1)$ time complexity using the memristive devices that are assembled in a crossbar array. This combination of binary/bipolar key memory and the mature controller in our architecture efficiently handles few-shot learning and classification of incoming unseen examples on the fly without the need for fine tuning the controller weights. For more details about the learning and inference phases see Supplementary Note 2.

**A new attention mechanism appropriate for HD geometry**. HD computing starts by assigning a set of random HD vectors to represent unrelated items, e.g., different letters of an alphabet[18].

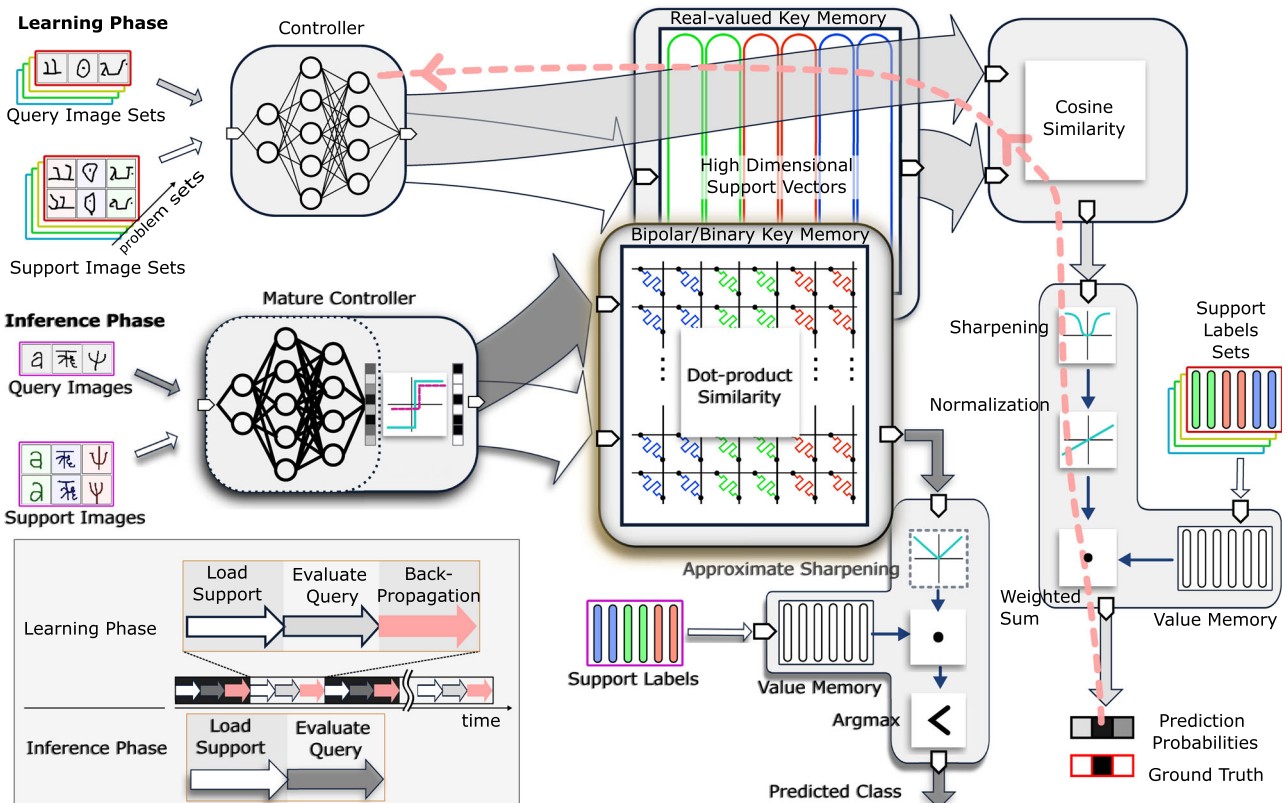

**Fig. 2 Proposed robust HD MANN architecture.** The learning phase of the proposed MANN involves a CNN controller which first propagates images in the support set to generate the HD support vector representations that are stored in the real-valued key memory. The corresponding support labels are stored in the value memory. For the evaluation, the controller propagates the query images to produce the HD vectors for the query. A cosine similarity module then compares the query vector with each of the support vectors stored in the real-valued key memory. Subsequently, the resulting similarity scores are subject to a sharpening function, normalization, and weighted sum operations with the value memory to produce prediction probabilities. The prediction probabilities are compared against the ground truth labels to generate an error which is backpropagated through the network to update the w eights of the controller (see pink arrows). This episodic training process is repeated across batches of support and query images from different problem sets until the controller reaches maturity. In the inference phase, we use a hardware-friendly version of our architecture by simplifying HD vector representations, similarity, normalization, and sharpening functions. The mature controller is employed along with an activation function that readily clips the real-valued vectors to obtain bipolar/binary vectors at the output of controller. The modified bipolar or binary support vectors are stored in the key memristive crossbar array (i.e., bipolar/binary key memory). Similarly, when the query image is fed through the mature controller, its HD bipolar or binary representation, as a query vector, is used to obtain similarity scores against the stored support vectors in the memristive crossbar array. The bipolar/binary key memristive crossbar approximates cosine similarities between a query and all the support vectors with the constant-scaled dot products in $\mathcal{O}(1)$ by employing in-memory computing. The results are weighted and summed by the support labels (in the value memory) after an approximate sharpening step and the maximum response index is output as the prediction.

The HD vector representation can be of many kinds (e.g., real and complex[24], bipolar[25], or binary[26]); however, the key properties are shared independent of the representation, and serve as a robust computational infrastructure[18,21]. In HD space, two randomly chosen vectors are quasi-orthogonal with very high probability, which has significant consequences for robust implementation. For instance, when unrelated items are represented by quasi-orthogonal 10,000-bit vectors, more than a third of the bits of a vector can be flipped by randomness, device variations, and noise, and the faulty vector can still be identified with the correct one, as it is closer to the original error-free vector than to any unrelated vector chosen so far, with near certainty[18]. It is therefore highly desirable for a MANN controller to map samples from different classes, which should be dissimilar in the input space, to quasi-orthogonal vectors in the HD feature space. Besides this inherent robustness, finding quasi-orthogonal vectors in high dimensions is easy and incremental to accommodate unfamiliar items[18]. In the following, we define conditions under which an attention function achieves this goal.

Let $\alpha$ be a similarity metric (e.g., cosine similarity) and $\epsilon$ a sharpening function. Then $\sigma$ is the attention function

$$\sigma(\mathbf{q}, \mathbf{K}_i) = \frac{\epsilon(\alpha(\mathbf{q}, \mathbf{K}_i))}{\sum_{j=1}^{mn} \epsilon(\alpha(\mathbf{q}, \mathbf{K}_j))}, \quad \alpha(\mathbf{q}, \mathbf{K}_i) = \frac{\mathbf{q} \cdot \mathbf{K}_i^T}{\| \mathbf{q} \| \| \mathbf{K}_i \|} \quad (1)$$

where $\mathbf{q}$ is a query vector, $\mathbf{K}_i$ is a support vector in the key memory, $m$ is the number of ways (i.e., classes to distinguish), and $n$ is the number of shots (i.e., samples per class to learn from). The key-value memory contains as many support vectors as $mn$. The attention function performs the (cosine) similarity comparison across the support vectors in the key memory, followed by sharpening and normalization to compute its output as an attention vector $\mathbf{w} = \sigma(\mathbf{q}, \mathbf{K})$ (see Methods). The cosine similarity has a domain and range of $\alpha : \mathbb{R}^d \times \mathbb{R}^d \to [-1, 1]$, where $\alpha(\mathbf{x}, \mathbf{y}) = 1$ means $\mathbf{x}$ and $\mathbf{y}$ are perfectly similar or correlated, $\alpha(\mathbf{x}, \mathbf{y}) = 0$ means they are perfectly orthogonal or uncorrelated, and $\alpha(\mathbf{x}, \mathbf{y}) = -1$ means they are perfectly anti-correlated. From the point of view of attention, two nearly

dissimilar (i.e., uncorrelated) vectors should lead to a focus closely to 0. Therefore, $\epsilon$ should satisfy the following condition:

$$\epsilon(\alpha(\mathbf{x}, \mathbf{y})) :\approx 0 \quad \text{when} \quad \alpha(\mathbf{x}, \mathbf{y}) \approx 0. \quad (2)$$

Equation (2) ensures that there is no focus between a query vector and a dissimilar support vector. The sharpening function should also satisfy the following inequalities:

$$\epsilon(\alpha) \geq 0 \quad (3)$$

$$\epsilon(\alpha_1) \leq \epsilon(\alpha_2) \quad \text{when} \quad \alpha_1 < \alpha_2 \quad \text{and} \quad \alpha_1, \alpha_2 > 0 \quad (4)$$

$$\epsilon(\alpha_1) \geq \epsilon(\alpha_2) \quad \text{when} \quad \alpha_1 < \alpha_2 \quad \text{and} \quad \alpha_1, \alpha_2 < 0, \quad (5)$$

Equation (3) implies non-negative weights in the attention vectors, whereas Eqs. (4) and (5) imply a strictly monotonically decreasing function on the negative axis and a strictly monotonically increasing function on the positive axis. Among a class of sharpening functions that can meet the above-mentioned conditions, we propose a soft absolute (softabs) function:

$$\epsilon(\alpha) = \frac{1}{1 + e^{-(\beta(\alpha - 0.5))}} + \frac{1}{1 + e^{-(\beta(-\alpha - 0.5))}} \quad (6)$$

where $\beta = 10$, as a stiffness parameter, which leads to $\epsilon(0) = 0.0134$. Supplementary note 3 shows the proof for softabs as a sharpening function that meets the optimality conditions.

As a common attention function, in various works[3,4,6,8] the cosine similarity is followed by a softmax operation that uses an exponential function as sharpening function ($\epsilon(\alpha) = e^{\alpha}$). However, the exponential sharpening function does not satisfy the above-mentioned conditions, and leads to undesired consequences related to the cost function optimization. In fact, when a query vector $\mathbf{q}$ belongs to a different class than some support vector $K_i$ and they are quasi-orthogonal to each other, then nevertheless $\mathbf{w}_i > 0$, where $\mathbf{w}_i = \sigma(\mathbf{q}, K_i)$. This eventually leads to a probability $\mathbf{p}_j > 0$ for class $j$ of support vector $i$. During model training, a well chosen cost function will penalize probabilities larger than zero for classes different from the query's class, and thus force the probability towards 0. This also forces $e^{\alpha}$ towards 0, or $\alpha$ towards $-\infty$. However, $\alpha$ only has a range of $[-1, 1]$ and the optimization algorithm will therefore try to make $\alpha$ as small as possible, corresponding to anticorrelation instead of uncorrelation. Consequently the softmax function unnecessarily leads to anticorrelated instead of uncorrelated vectors, as the controller is forced to map samples of different classes to those vectors.

The proposed softabs sharpening function leads to uncorrelated vectors for different classes, as they would have been randomly drawn from the HD space to robustly represent unrelated items (see Fig. 3a, b). It can be seen that the learned representations by softabs bring the support vectors of the same class close together in the HD space, while pushing the support vectors of different classes apart. This vector assignment provides higher accuracy, and retains robustness even when the HD real vectors are transformed to bipolar. Compared to the softmax, the softabs sharpening function effectively improves the separation margin between inter-class and intra-class similarity distributions (Fig. 3c, d), and therefore achieves up to 5.0%, 9.6%, 19.6% higher accuracy in 5-way 1-shot, 20-way 5-shot, and 100-way 5-shot problems, respectively (Fig. 3). By using this new sharpening function, our architecture not only makes the end-to-end training with backpropagation possible, but also learns the HD vectors with the proper direction. In the next sections, we describe how this architecture can be simplified, approximated, and transformed to a hardware-friendly architecture optimized for efficient and robust inference on memristive devices.

**Bipolar key memory: transforming real-valued HD vectors to bipolar.** A key memory trained with real-valued support vectors results in two considerable issues for realization in memristive crossbars. First, the representation of real numbers demands analog storage capability. This significantly increases the requirements on the NVM device, and may require a large number of devices to represent a single vector component. Second, a memristive crossbar which computes a matrix-vector product in a single cycle is not directly applicable for computing cosine similarities that are at the very core of the MANN architectures. For a single query, the similarities between the query vector $\mathbf{q}$ and all the support vectors in the key memory needs to be calculated, which involves computing the norm of $mn + 1$ vectors. An approximation strives to use the absolute-value norm instead of the square root[10], however it still involves a vector-dependent scaling of each similarity metric requiring additional circuitry to be included in the computational memory.

HD computing offers the tools and the robustness to counteract the aforementioned shortcomings of the real number representation by relying on dense bipolar or binary representation. As common properties in these dense representations, the vector components can occupy only two states, and pseudo-randomness leads to approximately equally likely occupied states (i.e., equiprobability). We propose simple and dimensionality-preserving transformations to directly modify real-valued vectors to dense bipolar and dense binary vectors. This is in contrast to prior work[11,13,14] that involves additional quantization, mapping, and coding schemes. In the following, we describe how our systematic transition first transforms the real-valued HD vectors to bipolar. Subsequently, we describe how the resulting bipolar HD vectors can be further transformed to binary vectors.

The output of the controller is a $d$-dimensional real vector as described in Section "A new attention mechanism appropriate for HD geometry". During the training phase, the real-valued vectors are directly written to the key memory. However, during the inference phase, the support vector components generated by the mature controller can be clipped by applying an activation function as shown in Fig. 2. This function is the sign function for bipolar representations. The key memory then stores the bipolar components. Afterwards, the query vectors that are generated by the controller also undergo the same component transformation, to generate a bipolar query vector during the inference phase. The reliability of this transformation derives from the fact that clipping approximately preserves the direction of HD vectors[27].

The main benefit of the bipolar representation is that every two-state component is mapped on two binary devices (see Supplementary Fig. 1). Further, bipolar vectors with the same dimensionality always have the same norm: $\| \hat{\mathbf{x}} \| = \sqrt{d}$, $\hat{\mathbf{x}} \in \{-1, +1\}^d$, where $\hat{\mathbf{x}}$ denotes a bipolar vector. This renders the cosine similarity between two vectors as a simple, constant-scaled dot product, and turns the comparison between a query and all support vectors to a single matrix-vector operation:

$$\alpha(\hat{\mathbf{a}}, \hat{\mathbf{b}}) = \frac{1}{d} \, \hat{\mathbf{a}} \cdot \hat{\mathbf{b}}^T \quad (7)$$

$$\mathbf{w} = \frac{1}{d} \, \hat{\mathbf{q}} \cdot \hat{\mathbf{K}}^T \quad (8)$$

As a result, the normalization in the cosine similarity (i.e., the product of norms in the denominator) can be removed during inference. The requirement to normalize the attention vectors is also removed (see the inference phase in Fig. 2).

**Binary key memory: transforming bipolar HD vectors to binary.** To obtain an even simpler binary representation for the

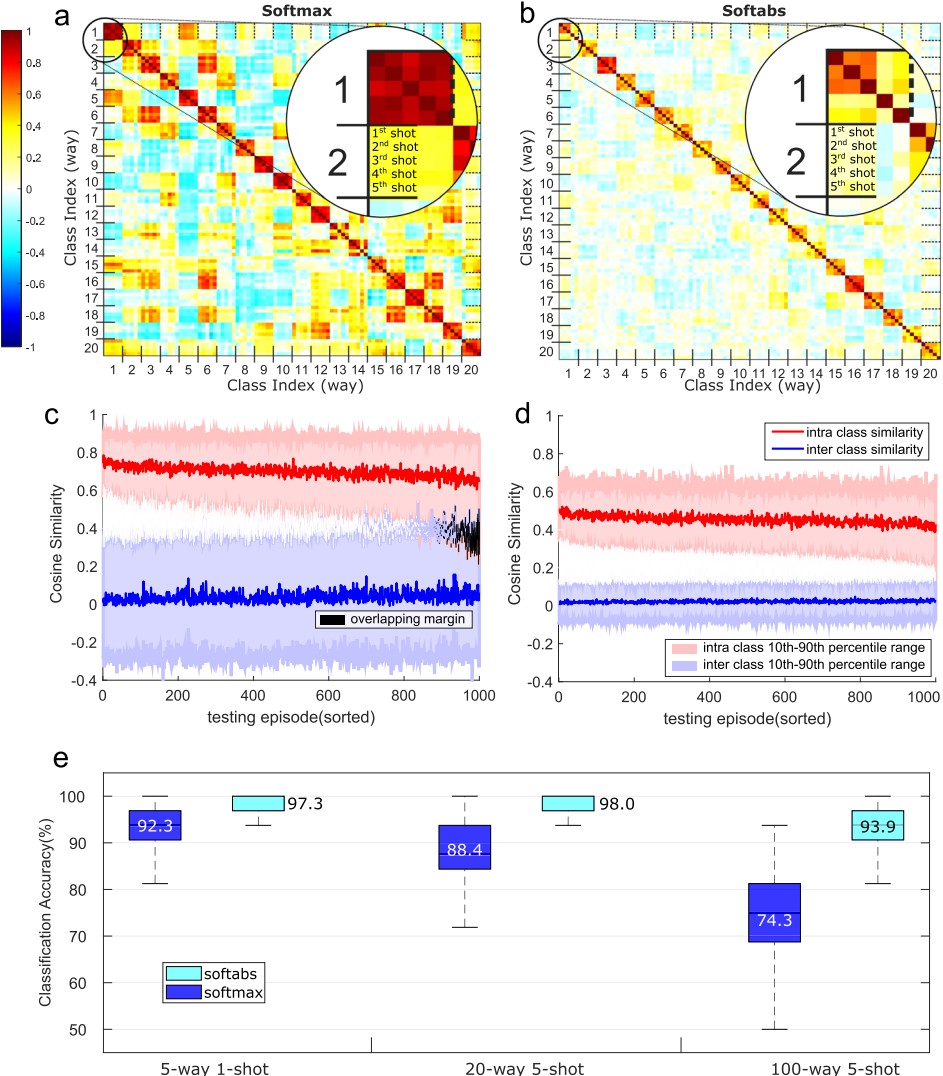

**Fig. 3 The role of sharpening functions.** The pairwise cosine similarity matrix from the support set of a single testing episode of 20-way 5-shot problem learned using the softmax (**a**) and the softabs (**b**) as the sharpening functions. Intra-class and inter-class cosine similarity spread across 1000 testing episodes in 20-way 5-shot problem with the softmax (**c**) and the softabs (**d**) as the sharpening functions (the episodes are sorted by the intra-class to inter-class cosine similarity ratio highest to lowest). In the case of the softmax sharpening function, the margin between 10th percentile of intra-class similarity and 90th percentile of inter-class similarity is reduced, and sometimes becomes even negative due to overlapping distributions. In contrast, the softabs function leads to a relatively larger margin separation (1.75×, on average) without causing any overlap. The average margin for the softabs is 0.1371, compared with 0.0781 for the softmax. (**e**) Classification accuracy in the form of a box plot from 1000 few-shot episodes, where each episode consists of a batch of 32 queries. The softabs sharpening function achieves better overall accuracy and less variations across episodes for all few-shot problems. The average accuracy is depicted in each case.

key memory, we used the following simple linear equation to transform the bipolar vectors into binary vectors

$$\hat{\mathbf{x}} = \frac{1}{2}(\hat{\hat{\mathbf{x}}} + 1) \qquad (9)$$

where $\hat{x}$ denotes the binary vector. Unlike the bipolar vectors, the binary vectors do not necessarily maintain a constant norm affecting the simplicity of the cosine similarity in Eq. (7). However, the HD property of pseudo-randomness comes to the rescue. By initializing the controller's weights randomly, and expanding the vector dimensionality, we have observed that the vectors at the output of the controller exhibit the HD computing property of pseudo-randomness. In case $\hat{\hat{\mathbf{x}}}$ has a near equal number of $-1$ and $+1$-components, after transformation with Eq. (9), this also holds for $\hat{x}$ in terms of the number of 0- and

1-components, leading to $\| \hat{\mathbf{x}} \| \approx \sqrt{\frac{d}{2}}$. Hence the transformation given by Eq. (9) approximately preserves the cosine similarity as shown below

$$
\begin{aligned}
\alpha(\hat{\mathbf{a}}, \hat{\mathbf{b}}) &\approx \frac{2}{d} \, \hat{\mathbf{a}} \cdot \hat{\mathbf{b}}^T \\
&= \frac{1}{2d}(\hat{\hat{\mathbf{a}}} + \mathbf{1}) \cdot (\hat{\hat{\mathbf{b}}} + \mathbf{1})^T \\
&= \frac{1}{2d}\left( \hat{\hat{\mathbf{a}}} \cdot \hat{\hat{\mathbf{b}}}^T + \underbrace{\sum_i \hat{\hat{\mathbf{a}}}_i + \sum_i \hat{\hat{\mathbf{b}}}_i}_{\approx 0} + d \right) \\
&\approx \frac{1}{2}\left( \frac{1}{d} \, \hat{\hat{\mathbf{a}}} \cdot \hat{\hat{\mathbf{b}}}^T + 1 \right) \\
&= \frac{1}{2}(\alpha(\hat{\hat{\mathbf{a}}}, \hat{\hat{\mathbf{b}}}) + 1),
\end{aligned}
\qquad (10)
$$

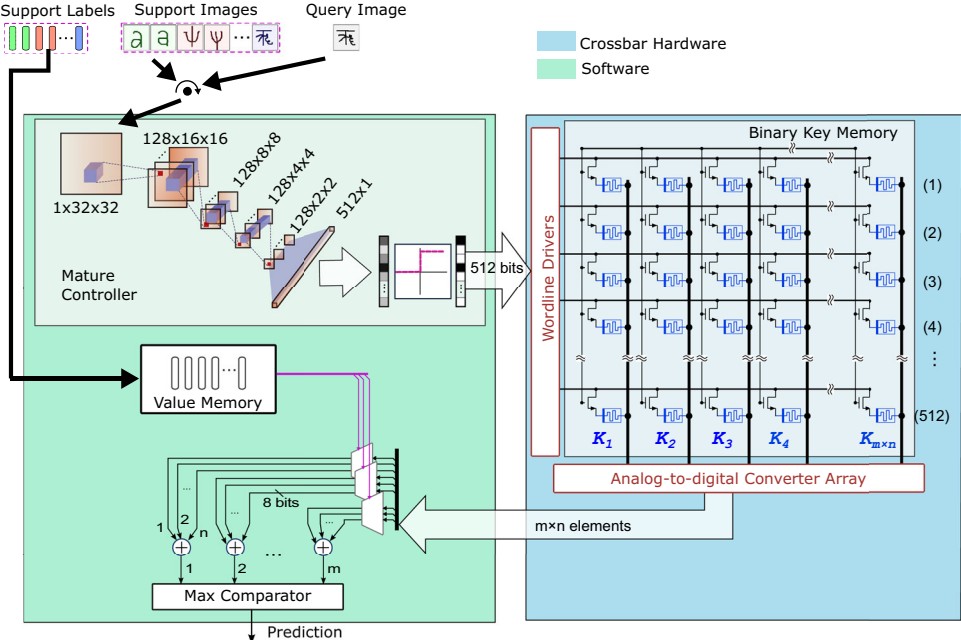

**Fig. 4 The MANN architecture with the binary key memory using analog in-memory computations.** The architecture is simplified for efficient few-shot inference: (1) The transformed HD support vectors are stored in a memristive crossbar array as the binary key memory; the query vectors are binarized too. (2) The cosine similarity ($\alpha$) between the input query vectors and the support vectors is computed through in-memory dot products in the crossbar using Eq. (10). (3) To further simplify the inference pipeline, the normalization of the attention vectors and the regular absolute sharpening function are bypassed. The accumulation of similarity responses belonging to the same support label in the value memory and finding the class with maximum accumulated response are implemented in software. The binary query/support vectors have 512 dimensions. $m$ and $n$ stand for "way" and "shot" of the illustrated problem respectively. A similar architecture with the bipolar key memory is shown in Supplementary Fig. 1.

where the approximation between the third and fourth line is attributed to the equal number of $-1$ and $+1$-components. We have observed that the transformed vectors at the output of the controller exhibit 2.08% deviation from the fixed norm of $\sqrt{\frac{d}{2}}$, for $d = 512$ (see Supplementary Note 1). Because this deviation is not significant, we have used the transformed binary vectors in our inference experiments. We also show that this deviation can be further reduced to 0.91% by training the controller to closely learn the equiprobable binary representations, using a regularization method that drives the HD binary vectors towards a fixed norm (see Supplementary Note 1). Adjusting the controller to learn such fixed norm binary representations improves accuracy as much as 0.74% compared to simply applying bipolar and binary transformations (See Supplementary Note 1, Table 2).

The proposed architecture with a binary key memory is shown in Fig. 4. Its major block is the computational key memory that is implemented in one memristive crossbar array with some peripheral circuitry for read-out. The key memory stores the dense binary representations of support vectors, and computes the dot products as the similarities thanks to the binary vectors with the approximately fixed norm. The value memory is at least $5 \times -100 \times$ smaller than the key memory, depending upon the number of ways, and stores sparse one-hot support labels that are not robust against variations (see Methods). Therefore, the value memory is implemented in software, where class-wise similarity responses are accumulated, followed by finding the class with maximum accumulated response (for more details on sum-argmax ranking see Supplementary Note 4).

**Experimental results.** Here, we present experimental results where the key memory is mapped to PCM devices and the similarity search is performed using a prototype PCM chip. We use a simple two-level configuration, namely SET and RESET conductance states, programmed with a single pulse (see Methods).

The experimental results for few-shot problems with varying complexities are presented. For the Omniglot dataset, a few problems have established themselves as standards such as 5-way and 20-way with 1-shot and 5-shots each[11,13,14,28–32]. There has been no effort for scaling to more complex problems (i.e., more ways/shots) on the Omniglot dataset so far. This is presumably due to the exponentially increasing computational complexity of the involved operations, especially the similarity operation. While the "complexity" of writing the key memory scales linearly with increasing number of ways/shots, the similarity operation (reading) has constant complexity $\mathcal{O}(1)$ on memristive crossbars. We have therefore extended the repertoire of standard Omniglot problems up to 100-way problem. For each of these problems we show the software classification accuracy for 32-bit floating point real number, bipolar and binary representations in Fig. 5a. To simplify the inference executions, we approximate the softabs sharpening function with a regular absolute function ($\epsilon_{\text{inference}}(\alpha) = |\alpha|$), which is bypassed for the binary representation due to its always positive similarity scores (see Supplementary Note 5). This is the only approximation made in the software inference, hence Fig. 5a reflects the net effect of transforming vector representations: a maximum of 0.45% accuracy drop (94.53% vs. 94.08%) is observed by moving from the real to the bipolar representation among all three problems. The accuracy drop from the bipolar to the binary is rather limited to 0.11% because both representations use the cosine similarity, otherwise the drop can be as large as 1.13% by using the dot product (see Supplementary Note 5). This accuracy drop in the binary representation can be reduced by using the regularizer as shown in Supplementary Note 1.

We then show in Fig. 5b the classification accuracy of our hardware-friendly architecture that uses the dot product to

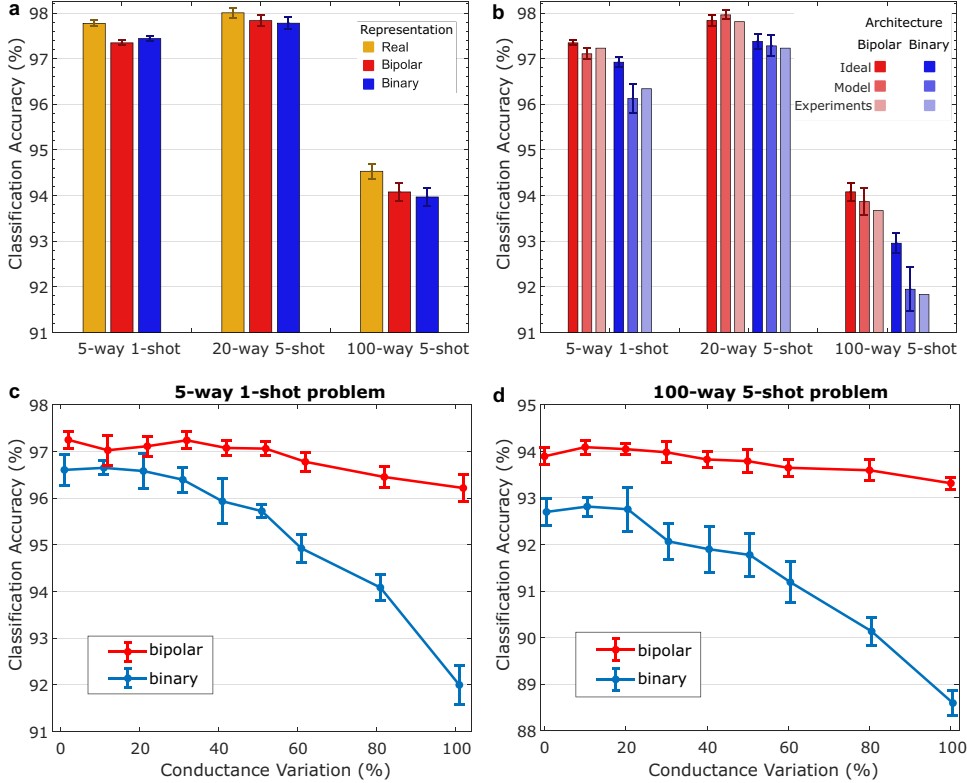

**Fig. 5 Experiments on Omniglot classification. a** Average software classification accuracy with the real, bipolar, and binary vector representations on three problems, each using the approximate sharpening function (i.e., the regular absolute), and the precise similarity function (i.e., the cosine) over 10 test runs each containing 1000 few-shot episodes (effectively 10,000 episodes); these capture the net effect of changing vector representations in software. **b** Classification accuracy results with the hardware-friendly inference architecture on an ideal crossbar without any PCM variations, a crossbar simulated with the PCM model (see Methods), and the actual experiments with the PCM devices (see Methods). The ideal and the PCM model simulations are conducted over 10 test runs each containing 1000 few shot episodes (effectively 10,000 episodes). The experiments were conducted over one test run containing 1000 episodes. **c** Classification accuracy as a function of percentage of device conductance variation in the PCM model with bipolar and binary as architectures for the 5-way 1-shot problem, and the (**d**) 100-way 5-shot problem. The error bars represent one standard deviation of sample distribution on either directions in all plots.

approximate the cosine similarity. The architecture adopts both binary and bipolar representations in three settings: (1) an ideal crossbar in the software with no PCM variations; (2) a PCM model to capture the non-idealities such as drift variability and read out noise variability (see Methods); (3) the actual experiments on the PCM hardware. As shown the PCM model accuracy is closely matched (±0.2%) by the PCM experiments. By going from the ideal crossbar to the PCM experiment, a maximum of 1.12% accuracy drop (92.95% vs. 91.83%) is observed for the 100-way 5-shots problem when using the binary representation (or, 0.41% when using the bipolar representation). This accuracy drop is caused by the non-idealities in the PCM hardware that could be otherwise larger without using the sum-argmax ranking as shown in Supplementary Note 4. For the other smaller problems, the accuracy differences are within 0.58%. Despite the variability of the key memory crossbar of the SET state at the selected conductance state (see Supplementary Note 6) our binary representations are therefore sufficiently robust against the deviations.

To further verify the robustness of the key memory, we conducted a set of simulations with the PCM model in Fig. 5c, d. We take the 5-way 1-shot and the 100-way 5-shot problems and compute the accuracy achieved by the architecture with respect to different levels of relative conductance variations. It can be seen that both the binary and the bipolar architectures closely maintain their original accuracies (with a maximum of 0.75%

accuracy drop) for up to 31.7% relative conductance deviation in the two problems. This robustness is accomplished by associating each individual item in the key memory with a HD vector pointing to the appropriate direction. At the extreme case of 100% conductance variation, the binary architecture accuracy degrades by 5.1% and 4.1%, respectively, for the 5-way 1-shot and the 100-way 5-shot problems. The accuracy of the bipolar architecture, with a number of devices doubled with respect to the binary architecture, degrades only by 0.93% and 0.58%, respectively, for the same problems. The bipolar architecture exhibits higher classification accuracy and robustness compared to the binary architecture, even with an equal number of devices, as further discussed in the next section, and illustrated in Supplementary Fig. 2.

We also compare our architecture with other in-memory computing works that use multibit precision in the CAM[11,13,14]. Our binary architecture, despite working with the lowest possible precision, achieves the highest accuracy across all the problems. Compared to them, our binary architecture also provides higher robustness in the presence of device non-ideality and noise (See Supplementary Note 7). Further, the other in-memory computing works cannot support the widely-used cosine similarity due to the inherent limitations of multibit CAMs. Therefore, we compare the energy efficiency of our binary key memory with a CMOS digital design that can provide the same functionality. The search energy per bit of PCM is 25.9 fJ versus 5 pJ in 65 nm digital (see Methods).

## Discussion

HD computing offers a framework for robust manipulations of large patterns to such an extent that even ignoring up to a third of vector coordinates still allows reliable operation[18]. This makes it possible to adopt noisy, but extremely efficient devices for off-loading similarity computations inside the key memory. Memristive devices such as PCM often exhibit high conductance variability in an array, especially when the devices are programmed with a single-shot (i.e., one RESET/SET pulse) to avoid iterative program-and-verify procedures that require complex circuits and much higher energy consumption[22]. While the RESET state variations are not detrimental because its small conductance value, the significant SET state variations of up to a relative standard deviation of 50% could affect the computational accuracy. Equation (11) provides an intuition about the relationship between deviations in the cosine similarity and the relative SET state variability:

$$\sigma(\Lambda) = \sqrt{\frac{2\alpha}{d}}\sigma_{rel} \qquad (11)$$

where $\Lambda$ denotes the result of the noisy cosine similarity operation, $\alpha$ denotes the cosine similarity value between the noise-free vectors, $d$ is the dimensionality of the vectors, and $\sigma_{rel}$ is the relative SET state variability. It states that the standard deviation of the noisy cosine similarity inversely scales with the square root of the vector dimensionality. See Supplementary Note 8 for the proof of Eq. (11). Supplementary Fig. 3 provides a graphical illustration of how the robustness of the similarity measurement improves by increasing the vector dimensionality. Hence, even with an extremely high conductance variability, the deviations in the measured cosine similarity are tolerable when going to higher dimensions. In the case of our experiments with $\sigma_{rel} = 31.7\%$ (see Methods), a dimensionality of $d = 512$ and a theoretical cosine similarity of $\alpha = 0.5$ (e.g., for uncorrelated vectors in the binary representations), the standard deviation in the measured cosine distance, $\sigma(\Lambda)$, is $\approx 0.015$.

The values for the standard deviations chosen in Fig. 5c, d are extremely high, yet the performance, particularly that of the bipolar architecture, is impressive. This could be mainly due to using twice the amount of devices per vector, as the bipolar vectors have to be transformed into binary vectors with dimensionality $2d$ to be stored on the memristive crossbar. However, when the binary and bipolar architectures use an equal number of devices (i.e., a bipolar architecture operating at half dimension of binary architecture), the bipolar architecture still exhibits lower accuracy degradation as the conductance variations are increased (see Supplementary Fig. 2). This could be attributed to better approximating the cosine similarity than the architecture with the transformed binary representation. Moreover, the softabs sharpening function is well-matched to the bipolar vectors that are produced by directly clipping the real-valued vectors, whereas there could be other sharpening function that favor learning the binary vectors.

Using a single nanoscale PCM device to represent each component of a 512-bit vector leads to a very high density key memory. The key memory can also be realized using other forms of in-memory computing based on resistive random access memory[33] or even charge-based approaches[34]. There are also several avenues to improve the efficiency of the controller. Currently it is realized as a deep neural network with four convolutional layers and one fully connected layer (see Methods). To achieve further improvements in the overall energy efficiency, the controller could be formulated as a binary neural network[35], instead of using the conventional deep network with a clipping activation function at the end. Another potential improvement

for the energy efficiency of the controller is by implementing each of the deep network layers on memristive crossbar arrays[36,37].

Besides the few shot classification task that we highlighted in this work, there are several tantalizing prospects for the HD learned patterns in the key memory. They form vector-symbolic representations that can directly be used for reasoning, or multi-modal fusion across separate networks[38]. The key-value memory also becomes the central ingredient in many recent models for unsupervised and contrastive learning[39–41] where a huge number of prototype vectors should be efficiently stored, compared, compressed, and retrieved.

In summary, we propose to exploit the robust binary vector representations of HD computing in the context of MANNs, to perform analog in-memory computing. We provide a novel methodology to train the CNN controller to conform with the HD computing paradigm that aims at first, generating holographic distributed representations with equiprobable binary or bipolar vector components. Subsequently, dissimilar items are mapped to uncorrelated vectors by assigning similarity-preserving items to vectors. The former goal is closely met by setting the controller–memory interface to operate in the HD space, by random initialization of the controller, and by real-to-binary transformations that preserve the dimensionality and approximately the distances. The quality of representations can be further improved by a regularizer if needed. The latter goal is met by defining the conditions under which an attention function can be found to guide the item to vector assignment such that the semantically unrelated vectors are pushed further away than the semantically related vectors. With this methodology, we have shown that the controller representations can be directed toward robust bipolar or binary representations. This allows implementation of the binary key memory on 256,000 noisy but highly efficient PCM devices, with less than 2.7% accuracy drop compared to the 32-bit real-valued vectors in software (94.53% vs. 91.83%) for the largest problem ever-tried on the Omniglot. The bipolar key memory causes less than 1% accuracy loss. The critical insight provided by our work, namely, directed engineering of HD vector representations as explicit memory for MANNs, facilitates efficient few-shot learning tasks using in-memory computing. It could also enable applications beyond classification such as symbolic-level fusion, compression, and reasoning.

## Methods

**Omniglot dataset: evaluation and symbol augmentation.** The Omniglot dataset is the most popular benchmark for few-shot image classification[23]. Commonly known as the transpose of the MNIST dataset, the Omniglot dataset contains many classes but only a few samples per class. It is comprised of 1623 different characters from 50 alphabets, each drawn by 20 different people, hence 32460 samples in total. These data are organized into a training set comprising of samples from 964 character types (approximately 60%) from 30 alphabets, and testing set comprising of 659 character types (approximately 40%) from 20 alphabets such that there is no overlap of characters (hence, classes) between the training set and the testing set. Before going into the details of the procedure we used to evaluate a few-shot model, we will present some terminology. A problem is defined as a specific configuration of number of ways and shots parameters. A run is defined as a fresh random initialization of the model (with respect to its weights), followed by training the model (i.e., the learning phase), and finally testing its performance (i.e., the inference phase). A support set is defined as the collection of samples from different classes that the model learns from. A query batch is defined as a collection of samples drawn from the same set of classes as the support set.

During a run, a model that is trained usually starts underfitted, at some point reaches the optimal fit and then overfits. Therefore it makes sense to validate the model at frequent checkpoint intervals during training. A certain proportion of the training set data (typically 15%) is reserved as the validation set for this purpose. The number of queries that is evaluated on a selected support set is called batch size. We set the batch size to 32 during both learning and inference phases.

One evaluation iteration of the model and update of weights in the learning phase, concerning a certain query batch, is called an episode. During an episode, first the support set is formed by randomly choosing $n$ samples (shots) from randomly chosen $m$ classes (ways) from training/validation/testing set. Then the query batch is formed by randomly choosing from the remaining samples from the

same classes used for the support set. At the end of an episode, the ratio between number of correctly classified queries in the batch versus the total queries in the batch is calculated. This ratio when averaged across the episodes is called training accuracy, validation accuracy, or testing accuracy depending on the source of the data used for the episode.

Our evaluation setup consists of a maximum 50,000 training episodes in the learning phase, the validation checkpoint frequency of once every 500 training episodes. At a validation checkpoint, the model is further evaluated on 250 validation episodes. At the end of training, the model checkpoint with the highest validation accuracy is used for the inference phase. The final classification accuracy that is used to measure the efficacy of a model is the average testing accuracy across 1000 testing episodes of a single run. This can be further averaged across multiple runs (typically 10) pertaining to different initializations of the model, since the model's convergence towards the global minimum of the loss function is dependent on the initial parameters.

To prevent overfitting and to gain more meaningful representations of the Omniglot symbols, we augmented the dataset by shifting and rotating the symbols. Specifically, every time we draw a new support set or query batch from the dataset during training, we randomly augment each image in the batch. For that we have two parameters $s$ and $r$ that we draw from a normal distribution with mean $\mu = 0$ and a certain standard deviation for every image, and shift it by $s$ and rotate it by $r$. We have found that a shifting standard deviation of $\sigma_s = 2.5$ pixels and a rotation standard deviation of $\sigma_r = \frac{\pi}{12}$ work well for $32 \times 32$ pixel images.

**The CNN as a controller for the MANN architecture.** For the Omniglot few-shot classification task, we design the embedding function $f(\mathbf{x}; \theta)$ of our controller as a CNN inspired by the embedding proposed in[14]. The input is given by grayscale 32 by 32 pixel images, randomly augmented by shifting and rotating them before being mapped. The embedding function is a non-linear mapping

$$f : \mathbb{B}^{32 \times 32} \to \mathbb{R}^d.$$

The CNN bears the following structure: two convolutional layers (each with 128 filters of shape $5 \times 5$), a max-pooling with a $2 \times 2$ filter of stride 2, another two convolutional layers (each with 128 filters of shape $3 \times 3$), another max-pooling with a $2 \times 2$ filter of stride 2, and finally a fully connected layer with $d$ units. The last fully connected layer defines the dimensionality of the feature vectors. Each convolutional layer uses a ReLU activation function. The output of the last dense layer directly feeds into the key memory during learning. During inference the output of the last dense layer is subjected to a sign or step activation (depending on the representation being bipolar or binary) before feeding into the key memory. The Adam optimizer[42] is used during the training with a learning rate of 1e-4. For more details of the training procedure refer to Supplementary Note 2.

**Details of attention mechanism for the key-value memory.** When a key, generated from the controller, belongs to the few-shot support set, it is stored in the key memory as a support vector during the learning phase and its corresponding label in the value memory as a one-hot support label. When the key corresponds to a query, it is compared to all other keys (i.e., the support vectors stored in the key memory) using a similarity metric. As part of an attention mechanism, the similarities then have to be transformed into weightings to compute a weighted sum of the vectors in the value memory. The output of the value memory represents a probability distribution over the available labels. The weightings (i.e., attention) vector has unit norm such that the weighted sum of one-hot labels represents a valid probability distribution. For an $m$-way $n$-shot problem with $s_i$ support samples ($i \in \{1, \ldots, mn\}$) and a query sample $x$, there is a parameterized embedding function $f_\theta$, with $p$ trainable parameters in the controller, that maps samples to the feature space $\mathbb{R}^d$, where $d$ is the dimensionality of the feature vectors. Hence, the set of support vectors $\mathbf{K}_i$, which will be stored in the key memory, and the query vector $\mathbf{q}$ are defined as:

$$\mathbf{K}_i = f_\theta(s_i), \quad \mathbf{q} = f_\theta(x)$$
$$\mathbf{K} \in \mathbb{R}^{mn \times d}, \quad \mathbf{q} \in \mathbb{R}^d, \quad \theta \in \mathbb{R}^p. \quad (12)$$

The attention mechanism is a comparison of vectors followed by sharpening and normalization. Let $\alpha$ be a similarity metric (e.g., cosine similarity) and $\epsilon$ a sharpening function (e.g., exponential function) with $\alpha : \mathbb{R}^d \times \mathbb{R}^d \to \mathbb{R}$, $\epsilon : \mathbb{R} \to \mathbb{R}$. Then,

$$\sigma(\mathbf{q}, \mathbf{K}_i) = \frac{\epsilon(\alpha(\mathbf{q}, \mathbf{K}_i))}{\sum_{j=1}^{mn} \epsilon(\alpha(\mathbf{q}, \mathbf{K}_j))}$$
$$\sum_{j=1}^{mn} \sigma(\mathbf{q}, \mathbf{K}_j) = 1 \quad (13)$$
$$\sigma : \mathbb{R}^d \times \mathbb{R}^{mn \times d} \to [0, 1]^{mn}$$

is the attention function for a query vector $\mathbf{q}$ and key memory $\mathbf{K}$, and its output is the attention vector $\mathbf{w} = \sigma(\mathbf{q}, \mathbf{K})$.

Similar support vectors to the query lead to a higher focus at the corresponding index. The normalized attention vector (i.e., $\sum_{i=1}^{mn} \mathbf{w}_i = 1$) is used to read out the value memory. The value memory contains the one-hot labels of the support samples in the proper order. A relative labeling is used that enumerates the support set. Using the value memory ($\mathbf{V} \in \mathbb{B}^{mn \times m}$), the output probability distribution and

the predicted label are derived as:

$$\mathbf{p} = \mathbf{w} \cdot \mathbf{V}$$
$$l_{\text{predicted}} = \arg \max_{i \in \{1, \ldots, m\}} \mathbf{p}_i. \quad (14)$$

Note that the output probability distribution p is the weighted sum of one-hot labels (i.e., the probabilities of individual shots within a class are summed together). We call this ranking sum-argmax that results in higher accuracy in the PCM inference experiments compared to a global-argmax where there is no summation for the individual probabilities per class. See Supplementary Note 4 for a comparison between these two ranking criteria.

**PCM model and simulations.** For the simulations of our architecture we use TensorFlow. A model implemented with the appropriate API calls can easily be accelerated on a GPU. We have also made use of the high-level library Keras, which is part of TensorFlow and enables quick and simple construction of deep neural networks in a plug-and-play like fashion. This was mainly utilized for the construction of the controller. For modeling the PCM computational memory, the low-level library API was used, since full control over tensors of various shapes and sizes had to be ensured. In order to model the most important PCM non-idealities, a simple conductance drift behavior has been assumed:

$$G(t) = G_{t_0} \cdot \left(\frac{t}{t_0}\right)^{-\nu} \quad (15)$$

where $\nu$ is the drift component and G means conductance after $t$ time since programming. Since we fit these parameters to our measurements, we can simply chose reference time $t_0 = 1$ sec so that Eq. (15) becomes

$$G(t) = G_0 \cdot t^{-\nu}. \quad (16)$$

We then introduce several parameters to model variations (see Supplementary Table 1). The variations are assumed to be of Gaussian nature. Our final model of the conductance of a single PCM device is the following:

$$G(t) = \mathcal{N}(0, \tilde{G}_r^2) + (G_0 \cdot \mathcal{N}(1, \tilde{G}_p^2)) \cdot t^{-\nu \cdot \mathcal{N}(1, \tilde{\nu}^2)}, \quad (17)$$

with $\mathcal{N}(\mu, \sigma^2)$ being the normal distribution with mean $\mu$ and standard deviation $\sigma$ and $\tilde{G}_r^2$, $\tilde{G}_p^2$, $\tilde{\nu}^2$ represent the variability in additive read noise, programming noise and drift respectively. Since we model a whole crossbar and time between successive query evaluations (estimated $1\mu sec$) of a batch is negligible compared to the evaluation time of the first query of the batch since programming (estimated $20s$), our simulation setup is simplified to batch-wise processing of multiple inputs (i.e. queries) to the crossbar, and thus we solve

$$\mathbf{I} = \mathbf{U} \cdot \mathbf{G}^T \quad (18)$$

in one step. Where U is read-out voltages representing the batch of query vectors, G is conductance value of the PCM array at evaluation time ($20s$) and I is the corresponding current values received for each query in the batch.

To derive the PCM model parameters, we SET 10,000 devices and measure their conductance over a time spanning 5 orders of magnitude. The distribution of the devices' conductance at two time instants is shown in Supplementary Fig. 4(a) and 4(b). The drift leads to a narrower conductance distribution over time, yet the relative standard deviation increases. In the interest of time scales used for the experiments, the PCM model simulations, uses $\sigma_{\text{rel}} = 31.7\%$.

In a second step, we fit a linear curve with offset $G_0$ and steepness $-\nu$ in a log-log regime of the measurements to each device measured (see Supplementary Fig. 4 (c) for a set of example measurements and their fitted curves). The mean values of all fitted $G_0$ and $\nu$ give us the parameters for the model. They are calculated to be 22.8 $\mu$S and 0.0715, respectively. Their relative standard deviation gives us the programming variability $\tilde{G}_p$ and drift variability $\tilde{\nu}$, which are calculated as 31.7% and 22.5% respectively. In order to derive the read-out noise $\tilde{G}_r$, we calculate the deviation of measured conductance from the conductance value obtained from the fit line for each point on the curve to retrieve the standard deviation. This gives us the standard deviation of the read-out noise as 0.926 $\mu$S.

**Experimental details.** For the experiments, we use a host computer running a Matlab environment to coordinate the experiments, which is connected via Ethernet with an experimental platform comprising two FPGAs and an analog front end that interfaces with a prototype PCM chip[22]. The phase-change memory (PCM) chip contains PCM cells that are based on doped-Ge₂Sb₂Te₂ (d-GST) and are integrated in 90 nm CMOS baseline technology. In addition to the PCM cells, the prototype chip integrates the circuitry for cell addressing, on-chip 8-bit analog-to-digital converter (ADC) for cell readout, and voltage- or current-mode cell programming. The experimental platform comprises the following main units: (1) a high-performance analog-front-end (AFE) board that contains the digital-to-analog converters (DACs) along with discrete electronics, such as power supplies, voltage, and current reference sources; (2) an FPGA board that implements the data acquisition and the digital logic to interface with the PCM device under test and with all the electronics of the AFE board; (3) a second FPGA board with an embedded processor and Ethernet connection that implements the overall system control and data management as well as the interface with the host computer.

The PCM array is organized as a matrix of 512 word lines (WL) and 2048 bit lines (BL). The PCM cells were integrated into the chip in 90 nm CMOS technology using the key-hole process[43]. The selection of one PCM cell is done by serially addressing a WL and a BL. The addresses are decoded and they then drive the WL driver and the BL multiplexer. The single selected cell can be programmed by forcing a current through the BL with a voltage-controlled current source. It can also be read by an 8-bit on-chip ADC. For reading a PCM cell, the selected BL is biased to a constant voltage of 300 mV by a voltage regulator. The sensed current, $I_{read}$, is integrated by a capacitor, and the resulting voltage is then digitized by the on-chip 8-bit cyclic ADC. The total time of one read is 1 $\mu$s. For programming a PCM cell, a voltage $V_{prog}$ generated off-chip is converted on-chip into a programming current, $I_{prog}$. This current is then mirrored into the selected BL for the desired duration of the programming pulse. The RESET pulse is a box-type rectangular pulse with duration of 400 ns and amplitude of 450 $\mu$A. The SET pulse is a ramp-down pulse with total duration of approximately 12 $\mu$s. This programming scheme yields a 0 S conductance for the RESET state and 22.8 × $10^{-6}$ S average conductance with 31.7% variability for the SET state (see Supplementary Table 1).

For the experiments on Omniglot classification, the MANN is implemented as a TensorFlow[44] model. For testing, the binarized, or bipolarized query and support vectors are stored in files. These are then accessed by the Matlab environment and either programmed onto the PCM devices (support vectors) or applied as read-out voltages (query vectors) in sequence.

When it comes to programming, in the case of binary representation, all elements of a support vector are programmed along a bit line so that binary 1 elements are programmed at SET state and binary 0 elements are programmed at RESET state. For bipolar representation, the support vectors are programmed along a pair of adjacent bit lines so that +1 elements are programmed to SET state at the corresponding wordline indices of the left bit line while −1 elements are programmed to SET state at the corresponding wordline indices of the right bit line. The rest of the locations in the PCM array are programmed to RESET state in bipolar experiments (see Supplementary Fig. 1). The relative placement of each support vector is arbitrarily determined for each episode independently.

Since the PCM devices are only accessible sequentially, we measure the analog read-out currents for each device separately using the on-chip ADC and compute the reduced sum along the bitlines digitally in order to obtain the attention values. The attention values are in turn stored in files again, which are accessed by the TensorFlow model to finalize the emulation of the key memory.

**Energy estimation and comparison**. To determine the level of energy efficiency of PCM key memory search operation with respect to alternative technologies, we develop a dedicated digital CMOS binary key memory in register transfer level (RTL). The CMOS binary key memory stores the support vectors in binary form, and computes dot product operation from a query binary vector to each of the support vectors in the key memory and outputs the resulting array of dot product values—similar to the functionality of the PCM hardware. The resources are allocated to the CMOS baseline in such a way that its throughput is equivalent to the PCM-based counterpart. The digital CMOS design is synthesized in a UMC 65 nm technology node using Synopsys Design Compiler. During post-synthesis simulation of 100 queries, the design is clocked at a frequency of 440 MHz to create a switching activity file. Then using Synopsys Primetime at the typical operating condition with voltage 1.2 V and temperature 25 °C, the average power values are obtained. Finally, the energy estimation is performed by integrating these average power values over time and normalizing it by diving by the number of key memory vectors and their dimensions, resulting in the search energy per bit of 5 pJ. The energy of PCM-based binary key memory is obtained by summing the average read energy per PCM device (2.5 fJ) and the normalized energy consumed by the analog/digital peripheral circuits (23.4 fJ).

## Data availability

The data that support the findings of this study are available from the corresponding authors upon request.

## Code availability

The code used to generate the results of this study is proprietary to IBM.

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

## Acknowledgements

This work was partially funded by the European Research Council (ERC) under the European Unions Horizon 2020 research and innovation program (grant agreement number 682675).

## Author contributions

A.R. defined the research question and direction. M.S. and G.K. conceived the methodology, and performed the experiments. M.L.G., G.C., L.B., A.S., and A.R. supervised the project. G.K. and A.R. wrote the manuscript with input from all authors.

## Competing interests

The authors declare no competing interests.
