## [Peer Review File · Nature Communications]

REVIEWER COMMENTS

Reviewer #1 (Remarks to the Author):

This work proposes a new MANN architecture that leverages the strengths of HD computing to generate the interface to key memory. The tested system includes a 4-layer CNN, that feeds into key memory 512-sized vectors which are encoded into binary HD representation, resulting in efficient search. The overall architecture is implemented in software, except for the key memory component, which is done in using PCM chip the authors published in previous work [22]. Only inference is run on PCM, training is done in software. The key difference in PCM implementation in author's previous work, relative to this work, is that work in [22] used wider vectors – of 10,000 bits, while this work uses 512 bit vectors.

Primary novelty beyond author's work in [22] is combining MANN with HD computing. In order to accomplish this, CNN's last layer had to be adapted, and the overall network had to be trained to produce 512-sized vectors at the output. As a part of this process, the authors propose using a softabs sharpening function and claim that this leads to uncorrelated vectors for different classes, however, they do not show theoretical proof of this. The original training was done using floating point, but two simple and previously published methods were provided to map the data into bipolar and binary representation, that is appropriate for PCM chip. The design is verified using Omniglot problem (100-way 5-shot).

Overall this is an interesting paper that illustrates benefits of HD computing for key memory implementation in MANN designs and as such would be of interest to the readers of this journal. The paper could be improved by implementing both training and inference in hardware, and by providing theoretical proofs to some of their claims in the paper. Current design has practically all of the implementation in software, and the component that is in hardware was already published recently.

Reviewer #2 (Remarks to the Author):

This manuscript presents an architecture that implements memory-augmented neural networks based on the use of a non-volatile memory with in-memory computing capabilities. The main elements of novelty here are the use of the Phase Change Memory elements, the attention mechanism and the hardware-friendly implementation of the mathematical model.

The amount of work and its complexity are impressive, the results are very interesting and the experimental setup is robust and convincing. I really appreciated this paper but I have to admit that reading it is quite difficult. Given the nature of the journal and its multidisciplinary audience, I think that the authors should try to improve the organization of the material. Probably, a preliminary Figure explaining the approach, before what is now FIG. 1 (the architecture), would help the readability, even if I understand that this hardly possible given the editorial limitations and guidelines. Maybe, it would be enough to rearrange a bit the organization of FIG. 1, to avoid confusion given by the superposition of elements describing the learning phase and the inference phase.

Reviewer #3 (Remarks to the Author):

The authors describe phase change memory implementation of few shot classification using high-dimensional computing. The key contributions are use of a softabs function, and implementation using phase change memory devices. The paper builds on past work published by the authors on in-memory hyperdimensional computing published in Nature Electronics in June 2020. My comments are listed below.

1. Few shot classification problem: The paper should present a review of the best results obtained on the fewshot classification problem by other authors. There should be a comparison between the accuracy of the proposed approach and prior approaches.

2. Energy consumption: The authors in their Nature Electronics paper list detailed energy consumption values. What are the advantages in energy consumption of the phase change memory implementation of the few shot image classification problem compared to other approaches including other in-memory approaches? A comparison of energy consumption would be useful.

3. CNN Controller: Is the CNN controller included in PCM or in software during inference? I assume it is implemented using software during training.

4. A comparison with an alternate controller such as LSTM could be useful. Comparison of the LSTM and CNN approaches would be useful.

Reviewer #1

This work proposes a new MANN architecture that leverages the strengths of HD computing to generate the interface to key memory. The tested system includes a 4-layer CNN, that feeds into key memory 512-sized vectors which are encoded into binary HD representation, resulting in efficient search. The overall architecture is implemented in software, except for the key memory component, which is done in using PCM chip the authors published in previous work [22]. Only inference is run on PCM, training is done in software. The key difference in PCM implementation in author's previous work, relative to this work, is that work in [22] used wider vectors – of 10,000 bits, while this work uses 512 bit vectors.

We would like to thank the reviewer for providing valuable opinion on our paper. It is true that the underlying core operation (dot product) of the hardware remains the same as in [22]. However, the key difference in this work compared to [22] is that we provide a methodology for algorithmic-hardware codesign targeting the MANNs in few-shot learning and inference tasks in the presence of noisy memristive hardware. More specifically, we show that we cannot *blindly* interface an associative memory used in [22] (similar to the key memory here) to a given MANN controller, as a very low accuracy (74%; see Fig 3.e) would result if the key memory vectors are expanded to high-dimensional (HD) vectors in a straightforward manner. Therefore, there is a need for a methodology to properly direct a powerful representation of the MANN that does matter for HD computing and robust classification with noisy memristive key memory. This methodology identifies the right set of tools, functions, and design choices, as summarized in the following:

- **Distance metric.** That is the first design choice. We choose the cosine distance metric instead of other metrics, such as Euclidean distance, L1-norm, and hyperbolic distance, because of its ease of approximation within an in-memory computing framework, due to the predominant use of dot product operations. The choice of cosine distance also allows end-to-end differentiability during training. We can then apply simple and dimensionality-preserving transformations to directly modify real-valued vectors to dense bipolar and dense binary vectors. This is in contrast to prior work [11,13,14] that involves additional quantization, mapping, and coding schemes.
- **Sharpening function.** Our methodology defines a set of conditions under which an optimal sharpening function can be found to interface the MANN controller with the HD key memory. We demonstrate that softabs yields a sharpening function that satisfies the optimality conditions, given the range of values of the cosine distance metric.
- **Regularization.** Our methodology introduces an additional regularizing term during training to drive the MANN controller to learn HD vectors with a fixed occupancy ratio. This regularization term directly learns binary vectors that are denser (reaching a fixed norm close to $\sqrt{d/2}$) than those vectors that are directly transformed from real-valued vectors. This improves the accuracy of dot product for the similarity computation on hardware (see Supplementary Note 1).
- **Dimensionality.** Our methodology flexibly explores the vector dimensionality to find a balance between the required robustness and compactness of vectors. By selecting 512 dimensions, we keep the standard deviation of cosine similarity measurements below 0.0015 for uncorrelated vectors, when the PCM devices operate in the typical SET state variability range (see Supplementary Figure 3).

- **Hardware-friendly approximations.** In our methodology, we introduce several approximations during inference that significantly reduce the hardware complexity, while retaining the classification accuracy. These include binarize/bipolarize activations, and approximations of the distance (similarity) measurement, as well as of the sharpening function.

In response: To clarify our contributions, we have updated abstract, introduction, and Section II.A. We have also added a new high-level figure as Fig 1 covering various aspects of our contributions.

Primary novelty beyond author's work in [22] is combining MANN with HD computing. In order to accomplish this, CNN's last layer had to be adapted, and the overall network had to be trained to produce 512-sized vectors at the output. As a part of this process, the authors propose using a softabs sharpening function and claim that this leads to uncorrelated vectors for different classes, however, they do not show theoretical proof of this.

As mentioned above, the choice of the sharpening function is one of the key components in our methodology for the MANNs. We admit the reason for its use and the justification of the choice of softabs as the sharpening function are not sufficiently well emphasized in the previous version of the paper. Thanks for raising this point.

In response: We have included a new Supplementary Note 3 showcasing the proof of softabs as a sharpening function that meets the optimality conditions.

The original training was done using floating point, but two simple and previously published methods were provided to map the data into bipolar and binary representation, that is appropriate for PCM chip.

We admit that the bipolarization method is as simple as in [27], but it is important to determine whether a MANN controller trained with the real representations would lead to a negligible loss in accuracy when a straightforward bipolarization (instead of the more complex transformations for in-memory computing in [11,13,14]) is suddenly introduced during inference. In fact, we had earlier employed a tanh-based activation function to guide the real values of the controller towards bipolar/binary states during training. We have then observed that with dimensions as high as 512, such activation functions are not required during training. Instead, applying the simple bipolarization followed by binarization is sufficient to maintain almost the same level of real representation accuracy. Note that we also provided a proof that explains why such binarization method works with noisy HD hardware (this has not been experimented and proved in any work so far): the proof states that the standard deviation of the noisy cosine similarity inversely scales with the square root of the vector dimensionality (Supplementary Note 8).

Moreover, to reduce the accuracy gap between the real and transformed binary representations, we introduce a regularization term that drives the MANN controller to directly learn dense binary representations with a fixed norm. Directly learning such dense binary representation improves accuracy as much as 0.74% compared to simply using bipolar and binary transformations (Supplementary Table II).

In response: We have updated Section II.D to further highlight the impact of the regularization term.

The design is verified using Omniglot problem (100-way 5-shot). Overall this is an interesting paper that illustrates benefits of HD computing for key memory implementation in MANN designs and as such would be of interest to the readers of this journal. The paper could be improved by implementing both training and inference in hardware, and by providing theoretical proofs to some of their claims in the paper. Current design has practically all of the implementation in software, and the component that is in hardware was already published recently.

We would like to thank the reviewer for the positive comments.

We agree that additional theoretical proofs are required. This was a significant shortcoming and we have addressed this in the revised manuscript. The new Supplementary Note 3 provides the proof of the optimality of the proposed sharpening function.

Unfortunately, there are no good reasons to perform training of the MANNs in the memristive hardware. In fact, we would either need a more complex hardware setup to compute the precise similarity required for training, or trade accuracy off by computing an approximate similarity on the crossbar array during training. Then there are further approximations that we introduce to simplify overall system implementation by exploiting the observation that the dataflow need not be differentiable for back propagation of error signals during inference. During training, the same set of approximations can no longer be applied, which would lead to significantly more complex hardware. Another reason that prevents us from doing training in hardware is the need to rewrite the whole array with a new support set at every episode, which would be prohibitive in terms of energy and endurance of PCM devices.

On the other hand, once a MANN controller is trained on a digital computer, in the inference phase such mature controller can be used with the PCM-based key memory for few-shot learning, where each row of the key memory will be written only once per training example (see the support set loading step in Supplementary Table II). This hybrid architecture meets the endurance requirements of PCM devices by programming (i.e., training) every PCM device only once for a given few-shot learning problem. In fact, for the few-shot learning hardware experiments reported in this paper, the PCM devices are programmed with a single RESET/SET pulse to avoid iterative program-and-verify procedures that require complex circuits and much higher energy consumption. Hence hardware implementation presented in this paper is really geared towards inference and not training. Hope the reviewer agrees with this reasoning.

Reviewer #2

This manuscript presents an architecture that implements memory-augmented neural networks based on the use of a non-volatile memory with in-memory computing capabilities. The main elements of novelty here are the use of the Phase Change Memory elements, the attention mechanism and the hardware-friendly implementation of the mathematical model. The amount of work and its complexity are impressive, the results are very interesting and the experimental setup is robust and convincing. I really appreciated this paper but I have to admit that reading it is quite difficult. Given the nature of the journal and its multidisciplinary audience, I think that the authors should try to improve the organization of the material. Probably, a preliminary Figure explaining the approach, before what is now FIG. 1 (the architecture), would help the readability, even if I understand that this is hardly possible given the editorial limitations and guidelines. Maybe, it would be enough to rearrange a bit the organization of FIG. 1, to avoid confusion given by the superposition of elements describing the learning phase and the inference phase.

We are truly thankful to the reviewer for the positive comments and suggestions for improvement. The concerns regarding the readability of the paper for a wider audience are well taken. In this context, we have given more thought into making the paper more accessible by modifying both text and graphical illustrations.

In response: As suggested, we added a simple introductory figure, as Fig. 1, conveying the main message of the paper. We have also updated abstract, introduction, and Section II.A to provide a better high-level view of our contributions.

Reviewer #3

The authors describe phase change memory implementation of few shot classification using high-dimensional computing. The key contributions are use of a softmax function, and implementation using phase change memory devices. The paper builds on past work published by the authors on in-memory hyperdimensional computing published in Nature Electronics in June 2020.

We thank the reviewer for the in-depth review of our paper and for the constructive comments. Indeed this work builds on top of our prior in-memory HD work.

However, the central contribution of this work is a framework and methodology to direct a neural network controller to encode information in a way that combines the richness of deep neural network representations with robust vector-symbolic manipulations of HD computing. We propose a methodology to achieve this goal, by having the CNN controller to assign quasi-orthogonal HD dense binary vectors to unrelated items in the key memory. In the first step, our methodology defines the proper choice of an attention mechanism, i.e., similarity metric and sharpening function, to enforce quasi-orthogonality (see Section II B). Next step is tuning the dimensionality of HD vectors between the last layer of CNN controller and the key memory. Finally, to ease inference, a set of transformation and approximation methods convert the real-valued HD vectors to the binary dense binary vectors (see Sections II C and II D). Or even more accurately, such binary vectors can be directly learned by our proposed regularization term (see Supplementary Note 1).

We felt that this higher-level objective was not adequately highlighted in our first submission and hence we have spent quite some effort on this in the revised submission. Hope this will be appreciated by the reviewer as well.

My comments are listed below.

1. Few shot classification problem: The paper should present a review of the best results obtained on the fewshot classification problem by other authors. There should be a comparison between the accuracy of the proposed approach and prior approaches.

The lack of comparison with other authors is indeed relevant. Although we provided a brief high-level qualitative comparison of our work to similar works in the literature, a comprehensive review was missing. Therefore, we have conducted a comparison of our work with other relevant meta-learning approaches targeting Omniglot few-shot learning tasks at three different levels of implementation details. First, we compare the accuracy of our model running entirely on 32-bit software against other software models. Note that we have extended the repertoire of standard Omniglot problems up to 100-way problems, as there is no work yet targeting that many ways. Then, we compare the accuracy of our model with the approximations and representations intended for the in-memory computing hardware against other models that report accuracy on hardware without noise. Finally, we compare the noise resiliency of our model with other in-memory computing works. In summary, our binary architecture, despite working with the lowest

possible precision, achieves the highest accuracy among all the other in-memory computing hardware using multibit CAMs. Compared to them, our binary architecture also provides higher robustness in the presence of device non-ideality and noise.

In response: We have included a new Supplementary Note 7 comparing the accuracy of our model against the state-of-the-art models in pure software, ideal hardware, and noisy hardware, by other authors on few-shot learning problems. We provided a summary of the results in Section II.E.

2. Energy consumption: The authors in their Nature Electronics paper list detailed energy consumption values. What are the advantages in energy consumption of the phase change memory implementation of the few shot image classification problem compared to other approaches including other in-memory approaches? A comparison of energy consumption would be useful.

Note that the other in-memory computing works [11,13,14] cannot support the widely-used cosine similarity due to the inherent limitations of multibit CAMs that have been mentioned in the introduction. Therefore, we have compared with a digital baseline that can provide the same functionality in similarity computation.

In response: We have added a new part in Method and updated Section II.E detailing the energy consumption of the proposed hardware against the equivalent digital CMOS.

3. CNN Controller: Is the CNN controller included in PCM or in software during inference? I assume it is implemented using software during training.

The CNN is implemented in software for both training and inference. Although it has been shown that the CNNs can be implemented in memristive crossbars [36,37] (mentioned in the third paragraph of discussion), in this work we focus on the hardware implementation of the explicit memory that represents the bottleneck for the MANN architectures (see Fig 4).

4. A comparison with an alternate controller such as LSTM could be useful. Comparison of the LSTM and CNN approaches would be useful.

We have chosen a CNN controller as Omniglot is the most popular benchmark for few-shot image classification [6,11,13,14,32]. One has to decide whether to use a recurrent or feedforward network as the controller based on the nature of problem. A recurrent controller such as LSTM has its own internal memory that can complement the larger explicit memory. We believe that LSTM controllers can reap similar benefits from our robust PCM-based key memory, leading to an expansion of the number of few-shot learning applications that can be supported. Depending on the complexity of the problem, hyperparameters of the LSTM controller at the interface with the key memory and loss functions may change. However, we reckon that the underlying hardware will remain essentially the same regardless of the controller type.

REVIEWERS' COMMENTS

Reviewer #1 (Remarks to the Author):

The updated version of the manuscript addressed my concerns well.

Reviewer #2 (Remarks to the Author):

The authors addressed my previous comments.

Reviewer #3 (Remarks to the Author):

Reviewer's comments have been addressed appropriately in the revised version.